# Frequency Response of Thermo-Optic Phase Modulators Based on Fluorinated Polyimide Polymer Waveguide

**DOI:** 10.3390/polym14112186

**Published:** 2022-05-27

**Authors:** Eun-Su Lee, Kwon-Wook Chun, Jinung Jin, Min-Cheol Oh

**Affiliations:** Department of Electronics Engineering, Pusan National University, Busan 46241, Korea; ensulee@pusan.ac.kr (E.-S.L.); kwchun@pusan.ac.kr (K.-W.C.); jinung@pusan.ac.kr (J.J.)

**Keywords:** fluorinated polyimide, optical phased array, phase modulator, polymer waveguide

## Abstract

Polymer waveguide phase modulators exhibit stable low-power phase modulation owing to their exceptional thermal confinement and high thermo-optic effect, and thus, have the merit of thermal isolation between channels, which is crucial for an optical phased array (OPA) beam scanner device. In this work, a waveguide phase modulator was designed and fabricated based on a high-refractive-index fluorinated polyimide. The propagation loss of the polyimide waveguide and the temporal response of the phase modulator were characterized. Moreover, the transfer function of the phase modulator including multiple poles and zeros was obtained from the measured frequency response. The polyimide waveguide modulator device demonstrated a fast response time of 117 μs for 1 kHz input signal, however, for 1 mHz step-function input, it exhibited an additional 5% phase change in 5 s.

## 1. Introduction

Optical waveguide phase modulators are key components necessary for beam scanning LiDARs, tunable lasers for 5G network, and optical current sensors [1,2,3,4,5]. Among these, the polymer waveguide thermo-optic phase modulator (TOPM) displays efficient phase modulation operated by low electric power owing to its high thermo-optic (TO) effect and excellent heat isolation property [6,7]. Therefore, when the polymer phase modulators are placed as an array with small pitches for an optical phased array (OPA) beam scanner device, thermal crosstalk between the adjacent channels is suppressed, enabling independent control of each channel [7,8].

The degree of integration of the photonic integrated circuits (IC) can be improved by incorporating high refractive index polymers through the enhanced confinement of the guided mode. Moreover, the smaller waveguide provides the faster response time of the TOPMs. Polyimides are polymers with high optical refractive indices and are widely used in various industrial applications, including passivation layers in semiconductor devices, owing to their thermal, mechanical, and chemical stability [9,10]. Among them, the fluorinated polyimide, containing C–F bonds in place of C–H bonds of the polymer, exhibits excellent optical transmittance and is useful for a substrate of OLED displays [11,12]. To date, several studies have investigated polymeric photonic IC devices comprising polyimides [13,14,15], and fluorinated polyimide materials with low transmission loss in the infrared region [16,17,18,19,20,21]. However, to the best of our knowledge, no systematic investigation has been performed to characterize the frequency response of the phase modulators made of fluorinated polyimide.

In this work, incorporating a fluorinated polyimide designed for OLED display, we designed a polymer waveguide TO phase modulator that was fabricated to characterize the optical loss and the modulation power efficiency. In addition, the frequency response for various input signal was measured to investigate the temporal response characteristics of the phase modulator. The transfer function of the device was obtained through a pole-zero analysis based on these results.

## 2. Design and Fabrication of Thermo-Optic Phase Modulators with Polymer Waveguides

A waveguide phase modulator array is required in an OPA beam scanner device that scans an output beam by adjusting the phase distribution of the output light that passes through the optical waveguide array. For OPA application, we investigated a TOPM array device consisting of 1 × 16 power-splitter, 16-channel phase modulators, and power combiners to produce 8-channel Mach–Zehnder interferometers (MZI), as illustrated in Figure 1a. The array device is useful for verifying the fabrication uniformity and thermal crosstalk between the phase modulators. A cross-sectional view of the waveguide is shown in Figure 1b, where a rib-structured waveguide core is above the lower cladding layer and a microheater is placed on the upper cladding. When electric power is applied to the heater, temperature change across the cross-section of the optical waveguide occurs, as shown in Figure 1c, resulting in a refractive index drop in the polymer and a corresponding phase change on the guided light.

Let the microheater cause a temperature change of $\Delta T_{h}$; then, the corresponding phase change is
(1)$$\Delta \phi =\frac{2\pi L}{\lambda }\cdot \frac{dN_{eff}}{dn}\cdot \frac{dn}{dT}\cdot \Delta T_{h}$$
where λ is the wavelength of light, L the length of the phase modulator heater, $dN_{eff}/dn$ the effective index change due to the refractive index change of polymer, and dn/dT is the TO coefficient of the polymer material. By defining the temperature change of the heater to produce Δϕ=π as $\Delta T_{h}^{\pi }$, the heating power for π phase shift, $P_{\pi }$ becomes,
(2)$$P_{\pi }=P_{ul}\cdot L\cdot \Delta T_{h}^{\pi }$$
where $P_{ul}$ is the heating power per unit length to increase the temperature by 1 °C, and this equals the rate of heat flow per unit length ($Q^{\prime }=dQ/dt$). Since $P_{ul}$ is inversely proportional to the heater length, $P_{\pi }$, which is proportional to the product of length and $P_{ul}$, remains constant regardless of the heater length. The rate of heat flow from the heater to the heat sink was calculated using Fourier’s law of heat conduction regarding the thickness and thermal conductivity of sandwiched materials. Inserting Equation (1) in Equation (2), the heating power for π phase shift becomes
(3)$$P_{\pi }=P_{ul}\cdot \frac{\lambda }{2}/\left(\frac{dN_{eff}}{dn}\cdot \frac{dn}{dT}\right)$$

The 2D finite-element method (FEM from OptoDesigner, Synopsys, Mountain View, CA, USA) was used to calculate $P_{\pi }$. First, the temperature change distribution was calculated considering the thermal conductivity and the TO coefficient of each material. The fluorinated polyimide material has a thermal conductivity of 0.2 W/mK and a TO coefficient of $-0.6\times 10^{-4}$/K. Then, the effective index change of the guided mode ($dN_{eff}/dT$) for the corresponding temperature distribution was found to calculate $P_{\pi }$ using Equation (3), and $\Delta T_{h}^{\pi }$ was obtained from Equation (2) [22]. $P_{\pi }$ was calculated to be 19.7 mW, and $\Delta T_{h}^{\pi }$ of the phase modulator was 11.8 °C.

A lumped thermal circuit model was used to analyze the response time of the TOPM. Thermal resistance ($R_{th}$) and thermal capacitance ($C_{th}$) of the layers constituting the device were used, then the time constant of the thermal response was obtained [23]. The lower thermal conductivity and capacitance of the device provides the faster response of the TOPM [24,25]. Compared to an ordinary polymer waveguide device consisting of thick polymer layers with poor thermal conductivity, 0.2 W/mK, in this work, silicon oxide with a thermal conductivity of 1.4 W/mK was selected as a lower cladding. Moreover, the thermal capacitance of the device was reduced by employing the high refractive index polyimide for the core layer so that the thickness of the entire polymer layer was reduced to 5 μm or less. In consequence, the thermal capacitance was reduced to less than one third of the previous study which had an 18 μm thick polymer layer [7]. To calculate the response time of the TOPM, FEM heat transfer simulation was performed. The transient thermal response was examined by applying temperature change in a step function of 11.7 °C corresponding to $\Delta T_{h}^{\pi }$. The rising and falling time for a 10–90% change in the core temperature was calculated as 108 μs.

The fluorinated polyimide was purchased from PI Advanced Materials Co. (Seoul, Korea), and the ZPU material was available from ChemOptics (Daejeon, Korea). The phase modulator was consisted of a lower cladding of silicon oxide with a refractive index of 1.4458, and a core of polyimide with a refractive index of 1.5613, and an upper cladding of ZPU430 polymer with a refractive index of 1.4300. A single-mode waveguide structure is needed to achieve high extinction ratio of the MZI device. When the refractive index contrast between the core and cladding materials is large, the rib waveguide could provide a single-mode waveguide with a relatively large core size, which is easier to fabricate. The rib-structured optical waveguide was designed using effective index method. The waveguide satisfies the single-mode condition with a core thickness of 3.3 μm and a width of 3 μm.

A schematic description of the fabrication process is drawn in Figure 2. A silicon wafer with a thermally grown oxide layer of 3 μm was used as a substrate. To improve the adhesion of fluorinated polyimide on a silicon-oxide surface, the wafer surface was treated in oxygen plasma followed by immersion in 0.1% (3-aminopropyl)triethoxysilane (APTES) solution for 30 s. Fluorinated polyamic acid solution in N-methyl-2-pyrrolidone (NMP) was then spin-coated at 4000 rpm onto the wafer surface for 60 s. The imidization process of the polyimide film was performed on a hotplate in a nitrogen environment. The temperature was gradually raised to 200 °C at a rate of 5 °C/min, stayed at 200 °C for 10 min for solvent evaporation, raised to 300 °C at a rate of 2.5 °C/min, then stayed at 300 °C for 30 min, resulting in a film thickness of 3.3 μm. Surface roughness of the film was measured using an atomic force microscope (AFM). The film exhibited good surface morphology and average surface roughness (*R_a_*) was measured as 1.18 nm. A waveguide pattern was formed on the cured polyimide film with a photoresist. The film was then etched by 2 μm in oxygen plasma to produce a rib waveguide with a remaining core layer of 1.3 μm. The upper cladding the ZPU polymer was spin-coated and cured with UV power of 9 mW/cm^2^ for 5 min in a nitrogen environment, and then the film was baked at 160 °C for 30 min. Microheater patterns were formed on the waveguide by photolithography and wet etching using Cr–Au of 10–100 nm thickness. After dicing the wafer, the end-facets were polished. A top view and a cross-sectional view of the fabricated device is shown in Figure 3b,c.

## 3. Temporal and Frequency Response of the Polyimide Phase Modulators

A DFB laser of 1550 nm wavelength was used to characterize the device for TE polarization. The optical mode profile of the fabricated waveguide was measured using a 40× objective lens and an infrared CCD camera. A polarization analyzer was placed between the objective lens and the CCD to verify the polarization states. The designed optical mode of the waveguide and the measured optical mode are shown in comparison in Figure 4. The mode field diameter (MFD) of the waveguide mode was measured using the mode of the single-mode fiber which has a MFD of 10.4 μm as the reference. MFD of the polyimide waveguide was measured as 3.6 × 3.7 μm^2^ (H × V), which was in good agreement with the design value of 3.4 × 3.6 μm^2^.

To characterize the loss of the device, instead of a cutback method that is time consuming due to repetitive end-facet preparation, in the present work, we prepared a series of waveguide patterns with different lengths and various bending angles, as shown in Figure 5a. Then, the insertion losses of the waveguides could be decomposed, at the same time, into propagation loss, bending loss, and coupling loss.

The total insertion loss of the waveguide can be written as a combination of propagation loss of the waveguide α, bending loss of the waveguide β, and the input–output coupling loss γ.
(4)$$L_{t}=\alpha \cdot l_{p}+\beta \cdot N+\gamma \cdot 2$$
where $l_{p}$ is the propagation length, and N the number of 90° bends. Insertion losses for a series of optical waveguide patterns with different lengths and bending structures form a matrix equation as,
(5)$$\left[\begin{array}{ccc} l_{p1} & N_{1} & 2\\ l_{p2} & N_{2} & 2\\ l_{p3} & N_{3} & 2\\ \vdots & \vdots & \vdots \\ l_{pn} & N_{n} & 2 \end{array}\right]\cdot \left[\begin{array}{c} \alpha \\ \beta \\ \gamma \end{array}\right]=\left[\begin{array}{c} L_{t1}\\ L_{t2}\\ L_{t3}\\ \vdots \\ L_{tn} \end{array}\right]$$

Here, $l_{p}$ and N are given for each waveguide pattern, and Equation (5) is an overdetermined system where the number of equations exceeds the number of unknown variables. The system is inconsistent, although a solution can be obtained with minimum error through the least-squares method using the algorithm (LSQR) provided in MATLAB software [26,27]. Based on the insertion loss results of Figure 5b, the propagation, bending, and coupling losses of the polyimide optical waveguide were estimated to be 0.91 dB/cm, 0.17 dB/90° bending, and 4.7 dB/coupling, respectively. The coupling loss is caused by the mode mismatch between the small waveguide core and ordinary single-mode fiber, and this could be reduced to 1.2 dB/coupling by adopting high-NA fibers.

Every two outputs of the 16-channel phase modulator array were combined to form eight Mach–Zehnder interferometers, as shown in Figure 1a, to investigate the characteristics of phase modulators. The device temperature was maintained at 50 °C with a deviation of less than 0.02 °C using a thermoelectric cooler (TEC) throughout the measurement. For a 10 Hz triangular signal with an amplitude to induce a phase modulation over 4π, an output signal was obtained, as shown in Figure 6a. For a 500 Hz square wave input signal adjusted to obtain maximum extinction ratio, the optical output was obtained, as shown in Figure 6b, where the 10–90% rise and fall time were 119 μs and 117 μs, respectively. The resistances of the 10-μm wide 3 mm long heaters of the 16-channel modulators were measured in the range of 267–300 Ω, and the $P_{\pi }$ of the 8-channel MZI was measured within 22.3–25.1 mW, as summarized in Figure 7. For TM polarization, $P_{\pi }$ was measured as 25.5 mW on average, which is slightly higher than that of the TE polarization. Variation in resistance occurred owing to the different lengths of the connecting lines to the contact pads. The $P_{\pi }$ was higher than the design results, which was due to the power consumption of the electrode connected to the contact pads. Simple calculation considering the width and length of the electrode explains that 84% of supplied power was consumed by the microheater.

The $P_{\pi }$ measure in this work is higher than the previous polymeric TOPMs [7]. There is a trade-off relation between the power consumption and the modulation speed for TOPMs. When the polymer waveguide becomes thinner for the faster speed, higher power consumption occurs because the heater gets close to the heat sink. In this study, we sacrificed the power consumption for the faster speed. However, the power consumption could be reduced by optimizing the device structure and by implementing air trenches around the core.

To characterize the temporal response of the phase modulator, a small amplitude signal introducing a phase change of π /10 was applied to the MZI phase modulator along with a DC bias. For various input signal frequencies, the amplitude and phase delay of the MZI output optical signal were measured to produce the Bode plot, as shown in Figure 8. By employing the circuit analysis technique, transfer functions of the MZI were formulated from the Bode plots as summarized in Table 1. When the transfer function was approximated as a first-order system, the primary pole showed up at 1.43 kHz, which was close to the 3 dB bandwidth obtained from the Bode plot. Then, the 10–90% rising time τ ($≅0.35/f_{3dB}$) given by the first-order RC circuit model became 265 μs. This was somewhat larger than the response time of 117 μs obtained from Figure 6b, and one can notice from Figure 8a that the signal amplitude decreased gradually and by a small amount at the low-frequency range below 100 Hz. This implied that the single-pole approximation was not sufficient to represent the experimental result.

To find a more accurate transfer function, it was necessary to employ the generalized Maxwell model including multiple poles and zeros [28]. A MATLAB function (tfest) was useful to find the transfer functions with many poles matching the experimental Bode plot. The transfer functions with three and five poles were obtained, as shown in Figure 9, and the equations are summarized in Table 1. The transfer function with five poles fitted very closely to the experimental results for the entire frequency range including the low-frequency region. In this way, we figured out the optimum transfer function of the TOPM, and it paved the way for the proper control of the polymer waveguide TOPM, including the complicated low frequency responses.

During the small-signal response measurement, especially for a low-frequency signal less than 10 mHz, we noticed the DC bias was hardly maintained, which resulted in an inaccurate measurement at this low frequency. Hence, the slow response was measured by applying a 1 mHz square wave input, as shown in Figure 10. The measured MZI output signal was converted to a phase change in 500 s. After the initial fast change in phase, it slowly increased about 5% of the total phase change in a few hundred seconds. Similar behavior was reported in previous studies of polymeric TOPMs [7,29].

Crosslinked polymers can be viewed as a mixture of flexible chains, and thermal energy constantly affects their intermolecular interactions [30]. Therefore, the slow phase change could have originated from the gradual rearrangement of the polymer network. In the case of polymer materials used for a mechanical system, both instantaneous elastic response and long-term viscous response were occurring during the process of reforming their molecular structure, well known as the viscoelasticity [28,31].

## 4. Verification of the Low Thermal Crosstalk between the Adjacent Polymer Modulators

The thermal crosstalk of the TOPM is an important issue to address for its application to an OPA device since it can lead to unstable phase control of each channel. To verify the low thermal crosstalk of the arrayed phase modulators, a heat transfer simulation was conducted for various channel distances, as shown in Figure 11. The bottom of the Si wafer was set as the heat sink, and an Au heater was placed on top of the upper cladding. A temperature increase was applied to the heater above one channel, and the temperature change in the adjacent channel was monitored. The temperature change of the adjacent channel was negligible for channel distances over 50 μm. About 3% of the temperature increase was observed for 50 μm distanced channel.

The low thermal crosstalk of the polymer phase modulator was verified by measuring several arrayed MZI devices with different arm distances (d), as illustrated in Figure 12a. In the presence of thermal crosstalk, the heat dissipated from one arm can affect the other and can induce a phase change in the other arm, leading to an increase of $P_{\pi }$. A polymer phase modulator fabricated using LFR polymer, a low-loss fluorinated polyacrylate polymer from ChemOptics, was used to confirm the minimum distance with low thermal crosstalk. Figure 12b shows the measured $P_{\pi }$ and extinction ratio (ER) of the arrayed MZIs. No significant change in $P_{\pi }$ was observed when the arm distance was over 50 μm. In the case of the 50 μm distanced arms, $P_{\pi }$ was increased by about 5%, which may not be critical for the OPA application.

## 5. Conclusions

Fluorinated polyimide material was used to fabricate optical waveguide phase modulators, and the temporal and frequency response characteristics were analyzed. The fluorinated polyimide waveguide exhibited a propagation loss of 0.9 dB/cm, and the average $P_{\pi }$ measured from phase modulator array was 24.0 mW. While the polymer phase modulators demonstrated in previous studies exhibited response time of several milliseconds, the current proposed phase modulator exhibited a fast response time of 117 μs which was mainly due to the reduction in the polymer layer thickness by incorporating a high refractive index polyimide core layer. The transfer function of the device was obtained by measuring the small signal frequency response and approximated with the simplest single pole system with the primary pole at 1.43 kHz corresponding to the time constant 265 μs. In the step-function response, we found an additional long-term phase change of about 5% occurring along with the initial fast response. This can be explained by the slow rearrangement of polymerized molecular network due to heat absorption, which caused the slow refractive-index change of the polymer film. The low thermal crosstalk of the polymer waveguide TOPM was verified. Through this study, an accurate understanding of the frequency response characteristics of the polyimide phase modulator was accomplished, and the low frequency response could be compensated by proper control of the polymeric phase modulators.

## Figures and Tables

**Figure 1 polymers-14-02186-f001:**
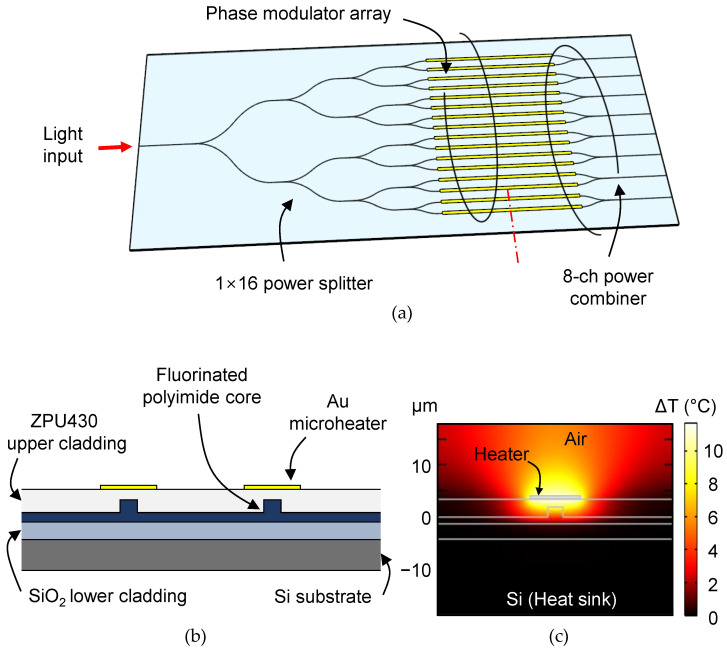
(**a**) Schematic of the polyimide waveguide thermo-optic phase modulator (TOPM) array which can be applied to an optical phased array (OPA) device. The TOPM array consists of a 1 × 16 beam splitter and 16 phase modulators (PM), and each of the two channels of the phase modulators is combined to form 8 Mach–Zehnder interferometers (MZI). (**b**) Cross-section of a MZI showing two cores of rib waveguides and microheaters on top of the upper cladding. (**c**) The heat distribution that occurs across the waveguide cross-section when the microheater is heated.

**Figure 2 polymers-14-02186-f002:**
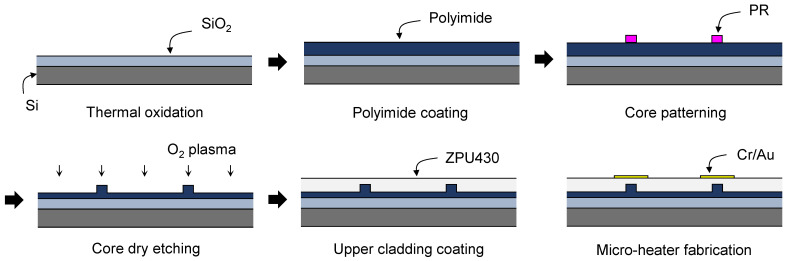
Schematic description of the device fabrication procedures.

**Figure 3 polymers-14-02186-f003:**
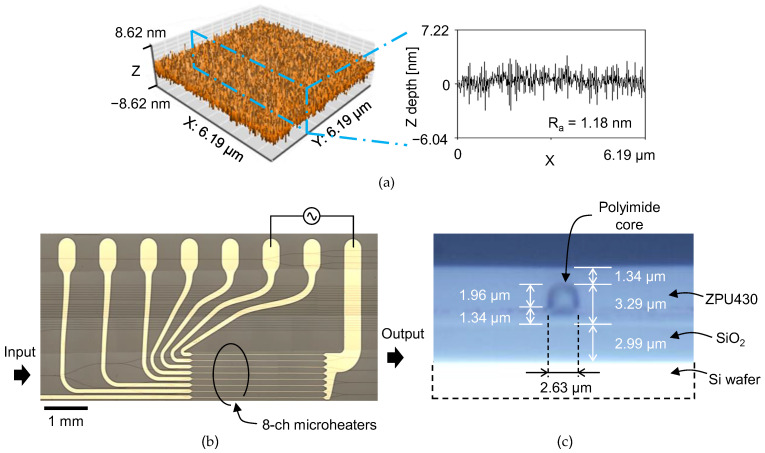
(**a**) Atomic force microscope image of the polyimide film and the cross-sectional plot showing depth profile. The average roughness ($R_{a}$) of the film surface was less than 2 nm. Mi-croscopic images of the fabricated phase modulator array: (**b**) top-view showing 1 × 8 power splitter, 8-ch phase modulator array, and 4-ch MZI array, which shows the half of the fabricated device and (**c**) waveguide cross-section formed by cleaving.

**Figure 4 polymers-14-02186-f004:**
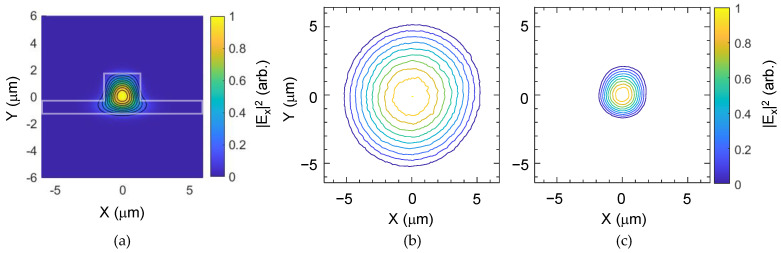
(**a**) Designed optical mode of the fabricated waveguide by using the finite difference method. Mode profile measurement results using a 40× objective lens and a CCD camera: (**b**) mode profile of a single-mode fiber used as the reference mode and (**c**) mode profile of the fabricated fluorinated polyimide waveguide. Note that the outermost contour line indicates a 1/*e*^2^ level of the total intensity

**Figure 5 polymers-14-02186-f005:**
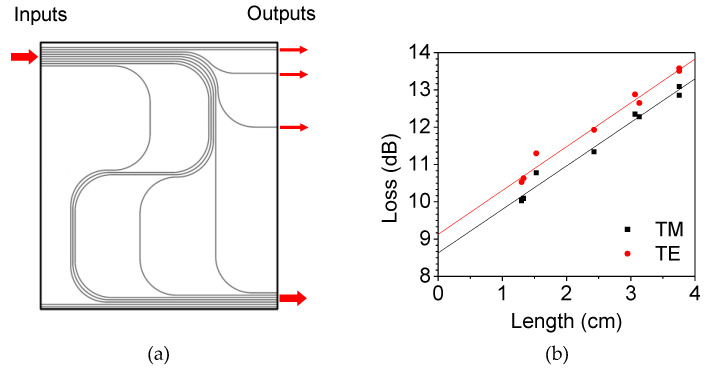
(**a**) Photomask layout of the waveguide patterns to measure the propagation loss, the bending loss, and the coupling loss of the fabricated waveguide. (**b**) The measured insertion losses plotted as a function of waveguide length of each waveguide pattern. From these results, the values of propagation, bending, and coupling losses were found by using the least-squares method from the overdetermined linear system.

**Figure 6 polymers-14-02186-f006:**
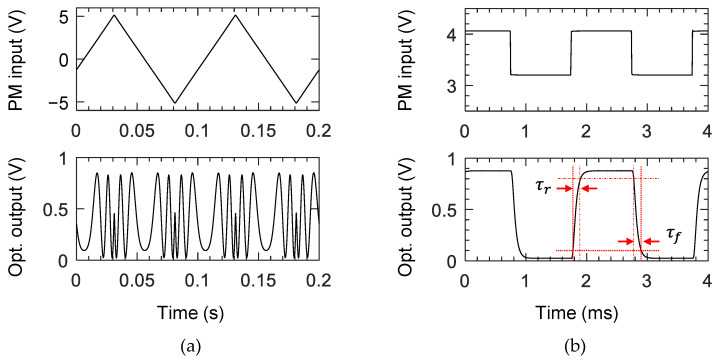
Intensity-modulated signal from Mach–Zehnder interferometer (MZI). (**a**) MZI output signal as a function of 10 Hz triangular signal input to induce a phase modulation over 4π, and (**b**) MZI output signal for a 500 Hz square wave input signal when the voltage levels were adjusted to obtain the best extinction ratio.

**Figure 7 polymers-14-02186-f007:**
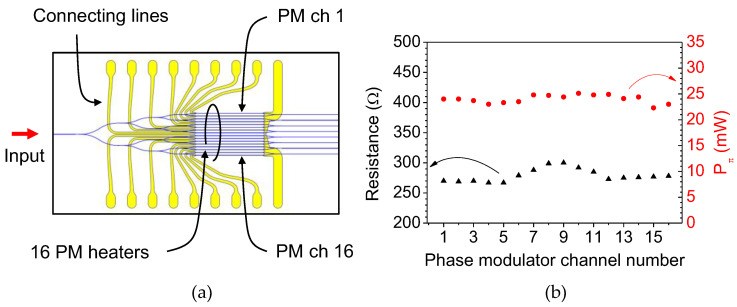
(**a**) Schematic of the 8-ch MZI array device. The PM microheaters on each channel were connected to the pads through connecting lines of different lengths, which led to variation in the heater re-sistance. (**b**) $R_{\pi }$ of the phase modulators and the resistance of the microheaters on each channel was measured to evaluate the fabrication uniformity of the array device.

**Figure 8 polymers-14-02186-f008:**
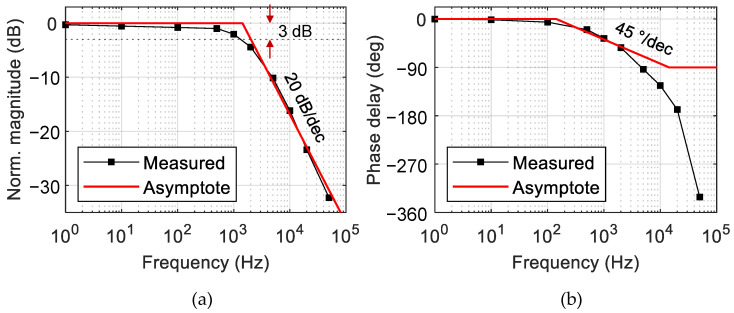
(**a**) Bode magnitude plot of the phase-modulation efficiency, and (**b**) phase delay Bode plot; the primary single pole was located near 1.43 kHz.

**Figure 9 polymers-14-02186-f009:**
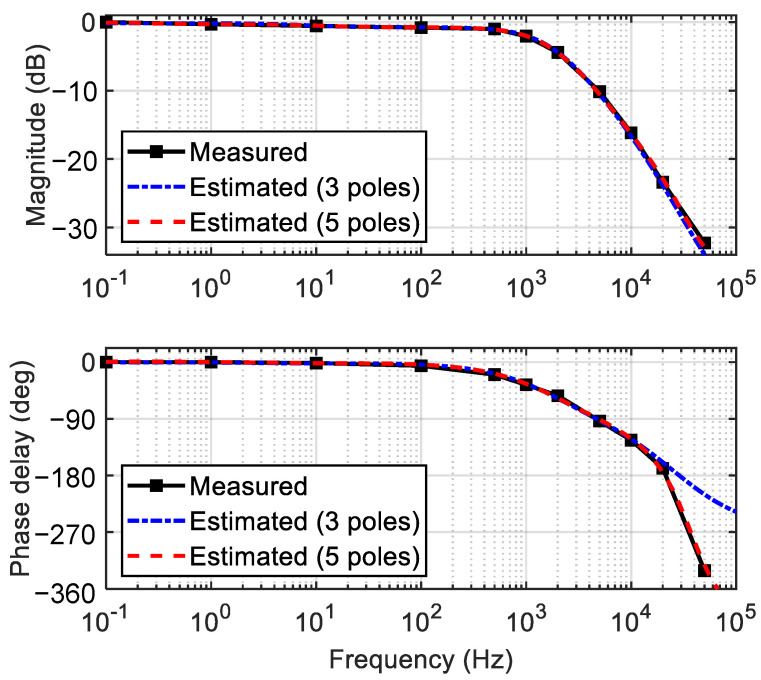
Bode plots reproduced by adopting the generalized Maxwell model incorporating three and five poles, which became close to the experimental results as the number of poles increased.

**Figure 10 polymers-14-02186-f010:**
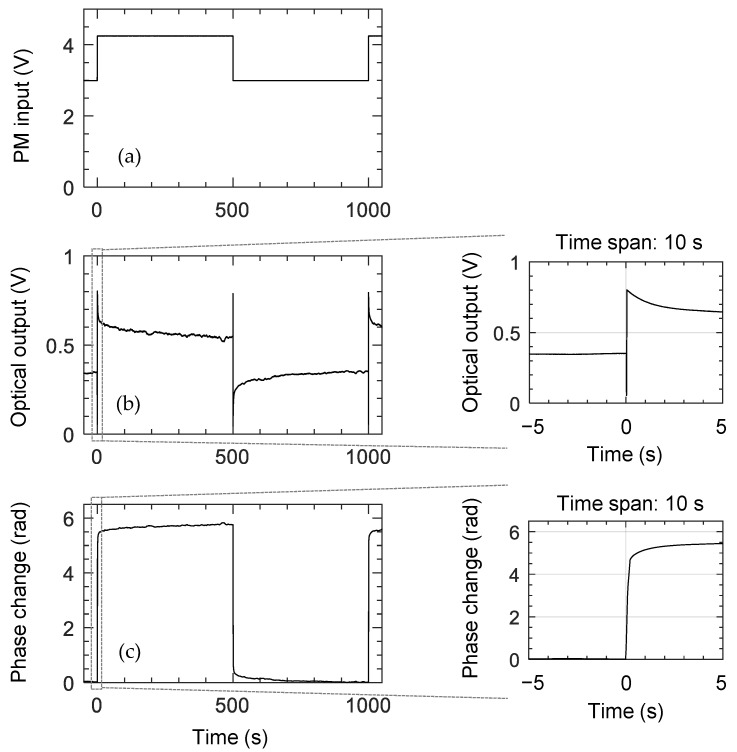
Phase change of Mach–Zehnder interferometer (MZI) modulator converted from the output intensity signal. (**a**) A 1 mHz square wave driving signal applied to the PM heater, (**b**) MZI output signal, which is converted to the phase change, and (**c**) the initial rapid response is enlarged in two graphs for a time span of 10 s. The phase changed rapidly at the beginning, then gradually increased by 5% with a slow time-constant due to viscous rearrangement of the polymer network.

**Figure 11 polymers-14-02186-f011:**
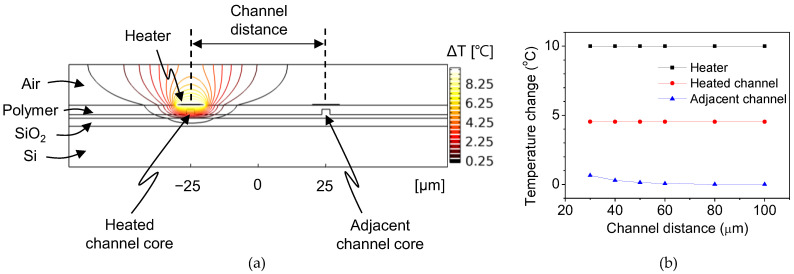
(**a**) Heat distribution around the heater and the waveguide channels when the channel distance was 100 μm. (**b**) Temperature change in the adjacent channel core compared to the temperature change of the heated channel for different channel distances.

**Figure 12 polymers-14-02186-f012:**
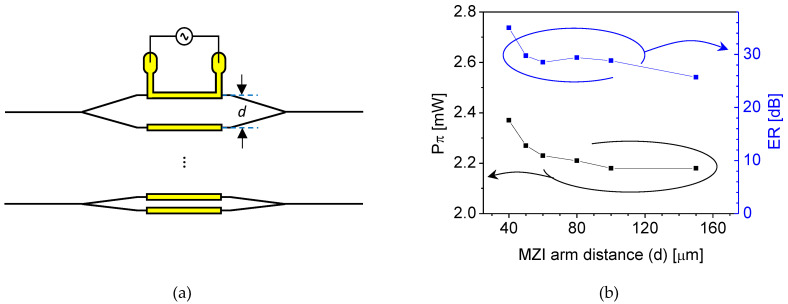
(**a**) Schematic diagram of the MZI array with different arm distances. (**b**) Measured $R_{\pi }$ and ex-tinction ratios (ER) of the phase modulators with 10-μm wide 3 mm long microheaters.

**Table 1 polymers-14-02186-t001:** Transfer functions in s-plane found by using a generalized Maxwell model to represent the experimental results depending on the number of poles (*N_p_*) incorporated.

N_p_	Transfer Function *H(s)*
1	$\frac{8603.6}{s+8970.5}$
3	$\frac{-5456.1(s+2.3023\times 10^{5})(s-52.224)}{\left(s-1.2427\times 10^{5})(s-1.0997\times 10^{4})(s-48.942\right)}$
5	$\frac{7818(s^{2}-2.7807\times 10^{5}s+5.8423\times 10^{10})(s-73.211)(s+1.7694)}{\left(s^{2}-2.9147\times 10^{5}s+4.6074\times 10^{10})(s-1.0785\times 10^{4})(s-69.763)(s+1.7121\right)}$

Here, *s = j ω*, units in rad/s.

## Data Availability

The data presented in this study are available on request from the corresponding author.

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
