# Peer review of "Frequency Response of Thermo-Optic Phase Modulators Based on Fluorinated Polyimide Polymer Waveguide"

_polymers, 2022, doi:10.3390/polym14112186_

Round 1
Reviewer 1 Report
The authors designed and fabricateda waveguide phase modulatorbased on a high-refractive-index fluorinated polyimide.Moreover, the low frequency response could be compensated by proper control of the polymeric phase modulators. The analysisprocess is substantial, which has practical value.The description of this work is detailed.However, this manuscript may be suitable for publication after a revision. I have several suggestions listed as follows,
Question 1:The ridge waveguide structure is chosen instead of rectangular waveguide structure when designing the TO waveguide modulator. The fabrication method of polymer rectangular waveguide is more convenient in experiment. What is the purpose of designing ridge waveguide?
Question 2:The Pπ of the 8-channel MZI was measured within 22.3–25.1 mW. Compared with some published results, the power consumption of this modulator is slightly higher. Does the author have methods to reduce the power consumption of the waveguide modulator?
Question3: Insertion losses plotted as a function of waveguide length of each waveguide patternaregiven in Figure 5, how to distinguish the TM modes and TE modes of output signal?
Question 4: The waveguide structure in Figure 11 is rectangular, but the ridge waveguide is designed by the authors, whether it has any effect on the simulation results.
Question 5:The author proposed polymer TO waveguide as optical phase modulator, however electro-optics(EO) polymer is commonly used as modulator. What are the advantages of the authors' design over EO waveguidemodulator?
Author Response
I would like to appreciate the reviewer for reviewing the manuscript and providing important comments to improve the quality of the manuscript. In accordance with the reviewer’s comments, I have prepared the responses described in this letter. The corrections are also reflected in the revised manuscript. Please see the attachment.

Reviewer 2 Report
The authors demonstrate a polymer waveguide optical phased array. M-Z interferometer has been introduced to study the phase modulation by thermo-optic effect. Polymer/silica hybrid waveguide structure has already been reported to improve the time response. The high-refractive index fluorinated polyimide has been used to increase the refractive index difference and restrain loss. Before the manuscript can be accepted, following issues are to be addressed.
1. In Fig. 5, the loss for TE and TM polarization has been experimentally studied. However, Section 3 Line 140, TE polarization is adopted as the input signal. Is there any characterization that has been implemented by TM polarization? What’s the performance of polarization dependence?
2. Section 3 Line 220-222, “To find more accurate transfer function, it was necessary to employ the Generalized Maxwell model including multiple poles and zeros. A MATLAB function (tfest) was useful to find the transfer functions with many poles matching the experimental Bode plot.” What’s the relationship between the final phase tuning performance and the fitting method?
3. In Section 4, what is the width of Au electrode? Will the width change of electrode have impact on the thermal crosstalk?
4. Fig. 11(a), the map heat distribution, the silica lower cladding has not been correctly noted. Besides, the ridge waveguide is said to be used in the manuscript, however, in Fig. 11(a), the rectangular waveguide is shown, why?
5. In Fig. 12(b), when investigating Pπ and ER as a function of d, only two electrodes have been considered, how about electrode with other width?
6. Fig. 6(a), Why the 10-Hz triangular signal is with an amplitude to induce the phase modulation over 4π? How about 2π?
Author Response

(The authors gave the same response as above.)
